# SUPERBLOOM: BLOOM FILTER MEETS TRANSFORMER

## ABSTRACT

We extend the idea of word pieces in natural language models to machine learning tasks on opaque ids. This is achieved by applying hash functions to map each id to multiple hash tokens in a much smaller space, similarly to a Bloom filter. We show that by applying a multi-layer Transformer to these Bloom filter digests, we are able to obtain models with high accuracy. They outperform models of a similar size without hashing and, to a large degree, models of a much larger size trained using sampled softmax with the same computational budget. Our key observation is that it is important to use a multi-layer Transformer for Bloom filter digests to remove ambiguity in the hashed input. We believe this provides an alternative method to solving problems with large vocabulary size.

## 1 INTRODUCTION

In natural language processing, one recent development, made popular by Wu et al. (2016) is to use a smaller sub-word vocabulary (Sennrich et al., 2016), or so called *word piece* model. In such a model, only frequent words and word pieces are kept in the vocabulary. Each word is then segmented as a sequence of word pieces. Both the input and the prediction are then represented in the smaller word piece space.

The word piece model has multiple benefits. Besides its generalizability and compact size, one crucial benefit is that we can afford to compute the full softmax loss on its much smaller vocabulary. This leads to more precise predictions, (measured e.g. using recall at $k$ for small values of $k$), compared to alternative approaches such as the sampled softmax method (Bengio & Sénécal, 2003; 2008) or the hierarchical softmax (Morin & Bengio, 2005). Word pieces have been shown to work well for natural language understanding (NLU) tasks. For example, the recent break-through of BERT (Devlin et al., 2018) uses a vocabulary of about 30K word pieces. The goal of this paper is to extend this idea to machine learning tasks where we have to model a large number of categorical values, which are represented by opaque ids (e.g. product ids, video ids) or named entities (e.g. Wikipedia or Knowledge Graph entities).

While word pieces are a natural way for breaking up words, it is unclear how this could be done for a set of arbitrary categorical values (referred to as vocabulary throughout the paper). We adopt a technique proposed by Serrà & Karatzoglou (2017) of using multiple hashing to reduce the vocabulary size while still keeping each entity identifiable. The idea is to map each id to multiple hash tokens in a smaller space, similarly to a Bloom filter (Bloom, 1970), and then embed and predict at the token, instead of the id, level. This way, the vocabulary is reduced to a much smaller size, similar to the effect of word pieces. However, hashing introduces much ambiguity due to random collisions. To solve this problem, we propose to use hashing in combination with a multi-layer Transformer (Vaswani et al., 2017), based on observations that the Transformer can disambiguate word meanings well using context. A hashed token can be viewed as a word piece with many different meanings. We hope that a Transformer model is also able to remove the ambiguity of hash tokens using the context, i.e. the set of other input tokens.

With these motivations, we build *Superbloom* in which we apply a Transformer model to the Bloom filter digest of the input. On the output side, the predictions are on the hashed tokens, similar to (Serrà & Karatzoglou, 2017). We demonstrate, through experiments, that Superbloom works well for tasks with a large vocabulary size – it can be efficiently trained and outperforms non-hashed models of a similar size, and larger models trained with sampled softmax with the same computational budget.

The key insight from our experiments is that the multiple-layer Transformer is effective for resolving the ambiguity in the hashed input, hence works particularly well for Bloom filter digests. For instance, we find that the model quality gap between a one layer and a twelve layer Transformer model is significantly larger when using Bloom filter digests, compared to that when the vocabulary is not hashed. This capability of the Transformer to "unhash" the Bloom digest is a key difference to earlier work on feature hashing (Weinberger et al., 2009) and multiple hashing (Serrà & Karatzoglou, 2017; Svenstrup et al., 2017; Daniely et al., 2017). In addition, we propose an efficient approximate inference algorithm, by applying "beam search" in the much smaller token space. Such fast inference is another useful property of Superbloom.

The Superbloom model is trained using BERT's masked language model task, i.e. on predicting hash tokens masked out from the input. Since the hashes have a much smaller vocabulary, this aspect is closely related to the error-correcting output codes (ECOC) (Dietterich & Bakiri, 1994; Berger, 1999)[1]. In the ECOC model, a multi-class classification is converted into many binary classification problems where redundancy is added to improve the robustness of the model. The prediction task of Superbloom can be viewed as reducing a large multi-class classification problem to a few smaller multi-class classification problems. However, unlike the ECOC models, we do not reduce the problem all way down to binary predictions. This might be the additional reason that Superbloom is able to achieve high accuracy.

## 1.1 RELATED WORK

Learning with a large vocabulary is a well-studied but still open research problem. Weinberger et al. (2009) proposed feature hashing which uses random hashing to reduce the input vocabulary size, and then learns embeddings for hashed ids in the smaller vocabulary. Several follow-up works propose to better resolve collisions by using multiple hashes: Svenstrup et al. (2017) proposed to learn a weighted sum of hashed embeddings; Shu & Nakayama (2018) used an unweighted sum, but proposed instead to learn the hash function itself; and Chen et al. (2018) proposed to learn both the hash function and the combiner, for which they use either a linear function or an LSTM. A key difference with the aforementioned work is that we do not resolve the hashing early at the input of the model; instead, we feed all hashed embeddings to the Transformer and let it learn to resolve the hashing collisions using the context. Our experiments show that multi-layer Transformer models indeed have the capacity to resolve hashing collisions while learning a high quality model.

Besides reducing the input space and memory usage, another set of related work focuses on dealing with large output vocabularies and improving training efficiency. A commonly used method is sampled softmax (Bengio & Sénécal, 2003; 2008) where for each gradient update, only a subset of the output vocabulary is considered. Another line of work is hierarchical softmax where classes are organized in clusters (Goodman, 2001) or in a tree structure (Morin & Bengio, 2005) to allow for efficient pruning of the output vocabulary. Through our experiments, we show that Superbloom, which allows us to train a full softmax on the hashed vocabularies, can lead to more accurate results than using sampled softmax on the larger output vocabulary. Serrà & Karatzoglou (2017) proposed to use Bloom filters as a general tool in deep models, for both the input and output. Our work demonstrates the efficiency of a multi-layer Transformer-like architecture to use contextual information to resolve hash ambiguity. Indeed, we show that shallow models, even with attention, fail.

## 2 SUPERBLOOM MODEL ARCHITECTURE

Given discrete sets $\mathcal{S}^I, \mathcal{S}^O$, representing respectively the input and output spaces (e.g. word tokens or entities), the goal is to model a function that maps a sequence of $n$ elements[2] in $\mathcal{S}^I$, to a sequence of probability distributions over $\mathcal{S}^O$. The space of probability distributions over a set $\mathcal{S}$ will be denoted by $\Delta(\mathcal{S}) = \{p \in \mathbb{R}_+^{|\mathcal{S}|} : \sum_{s \in \mathcal{S}} p_s = 1\}$.

The input and output entities are typically represented using embedding matrices $E^I \in \mathbb{R}^{|\mathcal{S}^I| \times d}$ and $E^O \in \mathbb{R}^{|\mathcal{S}^O| \times d}$, which map each entity to an embedding vector of dimension $d$. This makes training and inference expensive if the number of entities is very large. In order to reduce the model size and

---

[1]This connection is suggested by an anonymous reviewer.

[2]We assume a fixed sequence length for simplicity. This is also a useful assumption for practical implementation on a TPU, which requires fixed input dimensions.

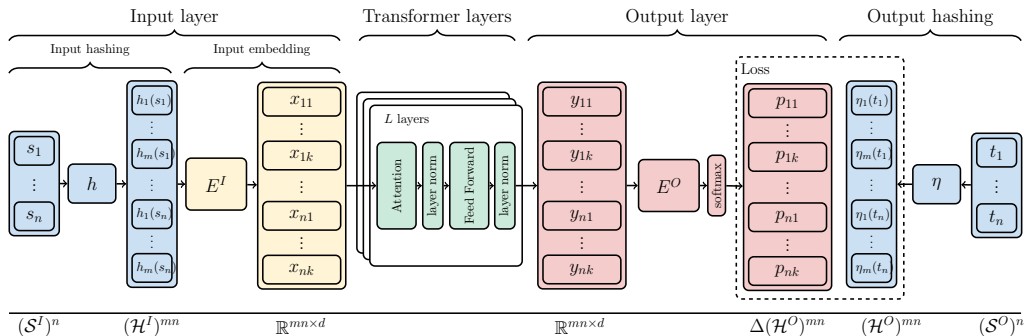

Figure 1: Superbloom model architecture

improve efficiency, we hash each element as follows. Given $m$ hash functions $h_j$, $j \in \{1, \ldots, m\}$, each element $s$ is represented by the hashes $(h_1(s), \ldots, h_m(s))$, which we refer to as a Bloom digest, given its similarity to the Bloom filter data structure. The set of values the hashes can take is much smaller than the original spaces $\mathcal{S}^I, \mathcal{S}^O$, which allows us to reduce the vocabulary size and thus the size of embedding matrices.

We decompose the Superbloom model architecture into $M = O \circ (T_L \circ \cdots \circ T_1) \circ I$, as illustrated in Figure 1: an input layer (Sec. 2.1) $I : (\mathcal{S}^I)^n \to \mathbb{R}^{mn \times d}$ which maps each item in the input sequence to $m$ embeddings of dimension $d$; $L$ transformer layers (Sec. 2.2) $T_i : \mathbb{R}^{mn \times d} \to \mathbb{R}^{mn \times d}$ which apply transformations in the embedding space; and an output layer (Sec. 2.3) $O : \mathbb{R}^{mn \times d} \to \Delta(\mathcal{H}^O)^{mn}$ mapping each embedding to a probability distribution. Since the model predicts distributions over $\mathcal{H}^O$ instead of $\mathcal{S}^O$, both training (Sec. 2.4) and inference (Sec. 2.5) need to be adapted accordingly.

## 2.1 INPUT LAYER $I : (\mathcal{S}^I)^n \to \mathbb{R}^{mn \times d}$

The input layer consists of $m$ hash functions[3] $h_j : \mathcal{S}^I \to \mathcal{H}^I$, $j \in \{1, \ldots, m\}$ and an embedding matrix $E^I \in \mathbb{R}^{|\mathcal{H}^I| \times d}$. The input sequence $(s_1, \ldots, s_n)$ is mapped to the sequence of embeddings $(E_{h_1(s_1)}, \ldots, E_{h_m(s_1)}, E_{h_1(s_n)}, \ldots, E_{h_m(s_n)})$. For ease of notation, we will write $x_{i,j} = E_{h_j(s_i)} \in \mathbb{R}^d$, and denote the sequence by $(x_{i,j})_{j=1,\ldots,m}^{i=1,\ldots,n}$. Throughout, we use subscripts $i, j$ to denote the $j$-th hash of the $i$-th element.

## 2.2 TRANSFORMER LAYERS $T : \mathbb{R}^{mn \times d} \to \mathbb{R}^{mn \times d}$

The Transformer is an attention-based model that was initially proposed for sequence transduction tasks, and that has been used in various other settings such as BERT. For the intermediate layers of Superbloom, we use the same architecture as the original transformer model (Vaswani et al., 2017), which we briefly summarize in Appendix A. Each transformer layer is a function $T : \mathbb{R}^{mn \times d} \to \mathbb{R}^{mn \times d}$ which maps a sequence of $mn$ embeddings in $\mathbb{R}^d$ to another sequence in the same space. We will denote by $(y_{i,j})_{j=1,\ldots,m}^{i=1,\ldots,n}$ the output sequence of the last transformer layer, where each $y_{i,j} \in \mathbb{R}^d$.

## 2.3 OUTPUT LAYER: $O : \mathbb{R}^{mn \times d} \to \Delta(\mathcal{H}^O)^{mn}$

Similarly to the input layer, we have $m$ hash functions $\eta_j : \mathcal{S}^O \to \mathcal{H}^O$, $j \in \{1, \ldots, m\}$ for the output space. We modify the original goal of predicting distribution over $\mathcal{S}^O$ to predicting distributions over $\mathcal{H}^O$, as follows. If $(y_{i,j})_{j=1,\ldots,m}^{i=1,\ldots,n}$ is the output of the last transformer layer, then the output layer $O$ maps each $y_{i,j}$ to

$$p_{i,j} := \sigma(y_{i,j}(E^O)^\top) \in \Delta(\mathcal{H}^O),$$

---

[3]The hash functions and their inverse mappings are randomly generated and stored as look-up tables. When generating the hash functions, we make sure that each hash bucket is evenly sized, and that there are no complete collisions.

where $E^O \in \mathbb{R}^{|\mathcal{H}^O| \times d}$ is an output embedding matrix, and $\sigma$ is the softmax function. Note that in some problems, the input and output spaces coincide, so it can be advantageous to use identical input and output hash functions, $h_j = \eta_j$, and the same embedding matrices $E^I = E^O$.

## 2.4 TRAINING

If the target sequence in $\mathcal{S}^O$ is $(t_1, \ldots, t_n)$, then the corresponding target sequence in $\mathcal{H}^O$ is $(\eta_j(t_i))_{j=1,\ldots,m}^{i=1,\ldots,n}$. We define the training objective[4] as

$$\sum_{i=1}^{n} \sum_{j=1}^{m} \ell(p_{i,j}, \eta_j(t_i)).$$

where $p_{i,j} \in \Delta(\mathcal{H}^O)$ are the model's output distributions, and $\ell : \Delta(\mathcal{H}^O) \times \mathcal{H}^O \to \mathbb{R}$ is a loss function, e.g. the cross-entropy loss. Note that we can pre-process the training data to map the elements in the original spaces $(\mathcal{S}^I)^n, (\mathcal{S}^O)^n$ to the hash spaces $(\mathcal{H}^I)^{mn}, (\mathcal{H}^O)^{mn}$, and training proceeds entirely in the hash spaces.

**Model size and efficiency** Compared to a model trained on the original space, the main advantage of Superbloom is a reduction in the size of the embedding matrices $E^I, E^O$. For instance, if a $\alpha$-to-one hashing is used (i.e., each hash bucket contains $\alpha$ elements), then $|\mathcal{H}| = |\mathcal{S}|/\alpha$ and the size of the input matrices is reduced by a factor $\alpha$. This not only reduces the memory cost, but may also improve the efficiency of gradient updates during training. Consider a cross-entropy loss, for each training example, all elements in the output space have a non-zero gradient due to the partition function in softmax, and thus the full matrix $E^O$ needs to be updated at each step, unless approximate methods such as sampled softmax (Bengio & Sénécal, 2003) are used. Our experiments (see Section 3.3) show that the cost of updating $E^O$ dominates that of training, and a reduction in vocabulary size allows us to significantly reduce training time without resorting to negative sampling.

---

**Algorithm 1** Approximate and exact inference in Superbloom

1: **Input:** Beam width $B$, a maximum iteration number $N$, model outputs $p_j \in \Delta(\mathcal{H}^O)$ and hash inverse look-up tables $\eta_j^{-1}$, for $j = 1, \ldots, m$.
2: For each $j$, sort $p_j$
3: **for** $b = B, \ldots, NB$ **do**
4:     Let $p_j^b$ be the $b$-th largest value in $p_j$.
5:     For each $j$, compute $S_j^b = \{s \in \mathcal{S}^O : p_j(\eta_j(s)) \geq p_j^b\}$.
6:     Score all candidates in $S^b = S_1^b \cup \cdots \cup S_m^b$. Let $s^\star = \arg\max_{s \in S^b} \gamma(s)$.
7:     **if** $\gamma(s^\star) \geq \sum_j \log p_j^b$ **then** break.       ▷ This guarantees $\gamma(s) \leq \gamma(s^\star)$ for all $s$.
8: **return** $s^\star$.

---

## 2.5 INFERENCE

For each position $i$ in the sequence, the model outputs $m$ distributions $(p_{i,j})_{j=1,\ldots m} \in \Delta(\mathcal{H}^O)^m$, and our goal is to use these distributions to rank the elements of $\mathcal{S}^O$. To simplify notation, we assume in this section that $i$ is fixed and will omit it by writing $p_j$ instead of $p_{i,j}$.

One simple way to rank items, as proposed by Serrà & Karatzoglou (2017), is to compute, for each $s$, $\gamma(s) := \sum_{j=1}^{m} \log p_j(\eta_j(s))$. When $\mathcal{S}^O$ is very large, this can be expensive, so instead of exhaustively scoring all items $s$, we propose an iterative beam-search, given in Algorithm 1 and illustrated in Figure 2, that can be used to compute the top-$k$ elements[5], either exactly, or approximately by fixing a maximum iteration number.

Let us fix a beam width $b$, and let $p_j^b$ be the $b$-th largest value of $p_j$, and let $S_j^b = \{s \in S^O : p_j(\eta_j(s)) \geq p_j^b\}$. In words, $S_j^b$ are elements whose hash is in the top $b$ values according to $p_j$.

---

[4]Note that unlike ECOC (Dietterich & Bakiri, 1994), the task is not to predict the individual bits in the output Bloom digest, but rather to predict (a probability distribution over) the index of the $m$ non-zero bits.

[5]For simplicity, we describe the algorithm for $k = 1$, but we apply it for larger $k$ in our experiments.

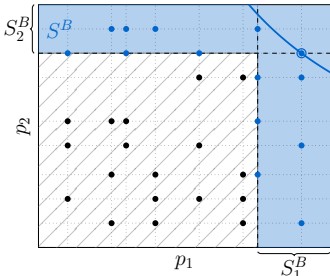 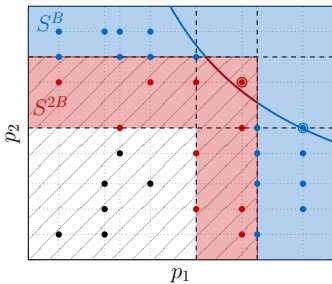

Figure 2: Illustration of approximate and exact inference, with a number of hashes $m = 2$, a four-to-one hashing scheme, and a beam width $B = 2$.

$S_j^b$ is obtained by pre-computing and storing inverse look-up tables[6] $\eta_j^{-1}$ for each hash function, and observing that $S_j^b = \cup_{h:p_j(h) \geq p_j^b} \eta_j^{-1}(h)$. This defines a set of candidates to score $S^b := S_1^b \cup \cdots \cup S_m^b$, and guarantees the following upper-bound: for all $s \notin S^b$, $\gamma(s) \leq \sum_j \log p_j^b$. If the best scored element $s^\star := \arg\max_{s \in S^b} \gamma(s)$ satisfies $\gamma(s^\star) \geq \sum_j \log p_j^b$, then we have a certificate that $s^\star$ is the best element over the entire set and the algorithm terminates. Otherwise, we increase the beam width and score a new set of candidates.

An example is illustrated in Figure 2 for $m = 2$, a beam width $B = 2$, and hash functions with $\alpha = 4$ (four elements share the same hash value along each dimension). The subset of candidates to score during the first iteration is highlighted in blue. The top element $s^\star$ is circled, and the solid line shows its $\gamma$ level set. In the left figure, the level set does not intersect the shaded area (unscored elements), thus we have a certificate that $s^\star$ is the exact maximizer. In the right figure, the level set does intersect the shaded area, so to find the exact maximizer, a second iteration is performed where the search region is extended (highlighted in red).

**Computational complexity**   Consider an $\alpha$-to-one hashing scheme. Sorting the vectors $p_j$ (line 2) costs $O(m|\mathcal{H}^O| \log |\mathcal{H}^O|)$. Each iteration consists of computing $S_j^b$ (line 5) then scoring candidates in $S^b$ (line 6) which costs $\mathcal{O}(m^2 B \alpha)$. The total cost for $N$ iterations is $O(m|\mathcal{H}^O| \log |\mathcal{H}^O| + m^2 N B \alpha)$ which can be significantly cheaper than scoring all candidates $\mathcal{O}(|\mathcal{S}^O|)$. For example, with the parameter setting described in Section 3.3, approximate inference is 10 times faster than exhaustive scoring. In Appendix B, we study the effect of the beam width on the quality of the model.

**Remark 1.** *While we describe a particular choice of ranking function $\gamma$, it is possible to generalize the algorithm to other ranking functions that are increasing, in a sense described in Appendix C.*

## 3   WIKIPEDIA ENTITY PREDICTION

We apply Superbloom to the Wikipedia entity prediction task, in which we use surrounding links on a Wikipedia page to predict a held-out link. This task is derived from the same data set as many NLU tasks, but uses entities instead of natural language. We believe this study is complementary to previous NLU models trained on Wikipedia, that focus on modeling language. Indeed, we show through examples that the model can learn entity relations well and demonstrates a strong use of contextual information.

The task needs to model about 5.3 million entity pages on Wikipedia. This vocabulary size is two orders of magnitude larger than in previous work that applies a Transformer model with full softmax loss (Devlin et al., 2018; Zhang et al., 2018; Sun et al., 2019). Other works, such as Zhang et al. (2019) and Soares et al. (2019), train a Transformer model with a large number of entities using sampled softmax, with either in-batch or in-example negative sampling. But as we shall show, sampled softmax, even with a large number of 128K negative samples, results in much worse quality.

---

[6]The cost of storing inverse look-up tables is dominated by that of storing embedding tables as a long as $m\alpha < d$ for an $\alpha$-to-one hashing scheme, since the inverse lookups have total size $O(m\alpha|\mathcal{H}^O|)$, while the embedding tables have size $O(d|\mathcal{H}^O|)$. This is always the case in our experiments.

## 3.1 TASK

We take all the entity pages on the website en.wikipedia.org. For each page, we obtain the URL links to other Wikipedia entity pages. We only use "raw" links, i.e. links that explicitly appear on the page. We obtain 5,281,889 pages and 462,588,415 links. Since the Wikipedia site usually removes duplicates of links on each page, the distribution of pages is rather long tail. For example, the top 100 most frequent pages represent only 3.8% of the total links, and the top 10% most frequent pages represent about 60% of the total links.

We hold out 10% random entity pages for testing. For the training data, we apply a masking similar to BERT – from each page, we take a random contiguous segment of entities, of length up to $n = 32$, and mask 15% of the segment. The task is then to predict the masked entities. We also apply the same input perturbation, where for the input, each masked out link is either replaced with a special [MASK] entity (with 80% probabilty), replaced with a random entity (with 10% probability), or left unchanged (with 10% probability). For evaluation, we hold out (i.e. replace with the [MASK] token) one random entity from a random segment on a test page. For quality evaluation, we use recall at $k$ metric (abbreviated as rec@$k$ below), which represents the chance the held out entity is in one of the top $k$ predictions.

## 3.2 MODEL

To apply Superbloom, we first create $m$ hash maps from entities to hash tokens with a given hash density $\alpha$. Each hash map is obtained by applying a random permutation to the vocabulary and map every consecutive $\alpha$ entities to the same token. This way we guarantee each hash token to have the same number of collisions $\alpha$.[7] Special tokens [CLS], [MASK], [SEP], are each mapped to $m$ tokens with no collisions. For example we create $[\text{MASK}_1], .., [\text{MASK}_m]$ tokens corresponding to [MASK].

We apply the hashing to the input and target, to map each entity to $m$ tokens as described in Section 2. We then apply the Transformer model to the input to predict the masked tokens. Unlike in BERT, we do not use position embeddings, in other words, we treat the input as a set instead of a sequence. Since the input and output spaces coincide, we use the same hash functions and the same embedding matrices in the input and output layer.

We carry out experiments on both the full vocabulary as well as a smaller subset consisting of the top 500K entity pages. On the smaller vocabulary, we are able to train a baseline model with large capacity, with no hashing and no sampling, which is useful for understanding the best achievable model quality.

We train all of our models on 16 Cloud TPUs. We use a batch size of $1024$ for experiments with full vocabulary and $4096$ for experiments with 500K vocabulary. All the experiments use the Adam optimizer (Kingma & Ba, 2014), and use a decreasing learning rate sequence with inverse square root decay, and initial learning rate 1e-4 for the full vocabulary and 2e-4 for the 500K vocabulary. All the experiments have been run for more than 1 million steps to reach near convergence.

## 3.3 SUPERBLOOM IS MORE ACCURATE

We experiment with two models of similar size: one is a baseline model (`baseline`) with full vocabulary of size $N$ equal to the number of entities; the other is a Superbloom model (`superbloom`) with a heavy 50 to 1 hashing. We set other hyper-parameters (such as the embedding dimension) so both models have a similar size. We also compare to a large model (`sampled-softmax`) trained using sampled softmax. Table 1 lists the hyper-parameters of each model. Recall that $\alpha$ denotes the number of collisions (1 if there is no hashing), $d$ the embedding dimension, $n_A$ the number of attention heads, $d_F$ the dimension of intermediate hidden layers, and $L$ the number of transformer layers. In all of our experiments, we use two hash functions for Superbloom models. Hence their vocabulary size is $2N/\alpha$.

---

[7]The procedure described here is for simplicity. If we are concerned with space, we may use some space efficient methods, for example a perfect hash function (Fredman et al., 1984).

| model | $\alpha$ | $d$ | $n_A$ | $d_F$ | $L$ | #parameters | #samples |
|---|---|---|---|---|---|---|---|
| baseline | 1 | 48 | 4 | 1024 | 12 | 248M | 5.3M |
| sampled-softmax | 1 | 512 | 8 | 2048 | 12 | 2.6G | 128K |
| superbloom | 50 | 768 | 12 | 3072 | 12 | 229M | 200K |

Table 1: Model parameters. "#samples" lists the number of samples in the softmax loss computation. For baseline and superbloom, since there is no sampling, this number corresponds to the full vocabulary, 5.3M and 200K, respectively. For sampled-softmax, we use 128K samples.

Table 2 shows the recall metrics of the models. For the Superbloom model, we set the beam width to $B = 20$ (our experiments suggest that it is sufficient to set $B = k$ in order to achieve the best rec@k metric, see Appendix B for details).

| model | rec@1 | rec@10 | rec@20 |
|---|---|---|---|
| baseline | 36.2% | 63.1% | 68.2% |
| sampled-softmax | 3.1% | 36.2% | 55.1% |
| superbloom | 51.1% | 72.3% | 76.5% |

Table 2: Recall metrics for different models.

The Superbloom model clearly outperforms, to a large extent, both the baseline and the sampled-softmax model. We note that the sampled-softmax model has much worse rec@$k$ than the other two models, and this gap is larger for smaller $k$. This is not surprising given the relatively small percentage (2.5%) of negative examples we can afford to sample.

While the Superbloom model performs well overall, there is a possibility that it devotes most of the embedding capacity to the top entities, so it loses accuracy on the less frequent entities. To test this, we plot the rec@1 value as a function of label frequency. In Figure 3, we show the mean rec@1 for every 10 percentile bucket in terms of the label frequency. We can observe that Superbloom is more accurate than the baseline in all the buckets. Another interesting phenomenon is that the most challenging labels are those in the 20 and 30 percentile. One possible reason is that they lack the higher predictability of the most frequent labels, and also the strong regularity of less frequent labels.

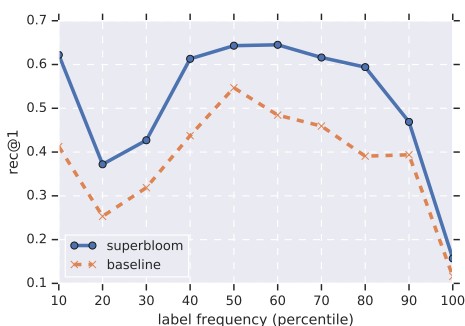

Figure 3: Rec@1 with respect to label frequency, starting from the most frequent labels.

Besides the high predictive accuracy, the prediction from the model shows strong semantic ability and context dependency. We show some examples of predictions in Figure 4 in Appendix E. In one set of examples, we pair "Copenhagen" with different entities, and observe that the predictions change accordingly, depending on the context. Another observation is that despite the heavy hashing, there are almost no unrelated entities in the top 10 predictions. The model even exhibits an ability to perform certain analogy tasks (without being trained on such tasks) – for example, given "Tunisia Tunis Thailand", it predicts "Bangkok" as the top result.

### 3.4 MULIT-LAYER TRANSFORMER IS IMPORTANT FOR SUPERBLOOM

Intuitively, given the large noise introduced by hashing, it is more important for Superbloom to use multiple attention layers in Transformer to "remove" the noise. To test this intuition, we run experiments with a smaller vocabulary size of the top 500K entity pages (about 60% of the links). On this smaller vocabulary size, we can afford to run a full softmax model with a larger embedding dimension.

| model | $\alpha$ | $d$ | $n_A$ | $d_F$ | $L$ | #parameters | rec@1 | rec@10 | rec@20 |
|---|---|---|---|---|---|---|---|---|---|
| baseline-l1 | 1 | 256 | 1 | 1024 | 1 | 123M | 51.0% | 70.4% | 75.5% |
| baseline-l12 | 1 | 256 | 8 | 1024 | 12 | 132M | 55.0% | 73.7% | 77.3% |
| superbloom-d256l1 | 20 | 256 | 1 | 1024 | 1 | 13M | 17.8% | 35.8% | 42.6% |
| superbloom-d384l1 | 20 | 384 | 1 | 1536 | 1 | 21M | 30.6% | 52.9% | 58.7% |
| superbloom-d256l12 | 20 | 256 | 8 | 1024 | 12 | 21M | 43.4% | 60.1% | 64.0% |

Table 3: Model parameters and recall metrics.

We consider different embedding dimensions and model complexity. Table 3 lists the model parameters as well as the recall metrics for each model. We observe that for the baseline models, the quality difference is small between models of different complexity. For example, rec@1 of baseline-l12 (55.0%) is about 8% better than baseline-l1 (51.0%). Since a one layer Transformer is close to a bag-of-words (BOW) model, one may argue that it may be unnecessary to use a Transformer in this case – instead one can use a larger dimension BOW model to achieve a similar accuracy.

However, for Superbloom models, the quality improves significantly with more layers. When increasing the number of layers from 1 (superbloom-d256l1) to 12 (superbloom-d256l12), rec@1 increases from 17.8% to 43.4%. The multi-layer model also performs much better than the single layer model with the same size (superbloom-d384l1). Note that previous work on hashed vocabularies relies on BOW models, which are less expressive than even a single-layer transformer. This highlights one of our key observations that multi-layer Transformer models are more effective for working with hashed vocabularies.

## 4 EXPERIMENTS ON NATURAL LANGUAGE DATA

In this section, we apply Superbloom to natural language data. We consider a large vocabulary that contains frequent unigrams and bigrams and use it to tokenize the text, then apply a Bloom filter to reduce the vocabulary size. We show that despite high hash collisions, the model can achieve high accuracy on natural language data. Since many named entities appear in the large vocabulary, we observe that the model seems to make better predictions of named entities than the BERT model.

While each hash id can be regarded as a word piece in an NLU model, there are important differences between hash ids and word pieces. First, hashing causes random collisions, while wordpiece tokenization can be viewed as a special hashing scheme based on the spelling – there is often coherence between words that share a word piece. As suggested by the experiments in Appendix D, random hashing with Superbloom digests may outperform coherent hashing. In addition, as every token in the large vocabulary is hashed, we do not have unambiguous anchors (such as the exact word pieces) to help bootstrap the disambiguation process. Despite these differences, our experiments suggest that even with high hashing collision $\alpha = 40$, the Transformer is capable of resolving, or unhashing, the Bloom filter digest effectively and produces highly accurate predictions and meaningful embeddings.

We construct a vocabulary of size 1M by taking the union of standard BERT word piece vocabulary ($\sim$ 30K) with the most frequent unigrams and bigrams, and follow the same procedure in BERT to create training examples. For Superbloom, we apply random hash maps to the 1M vocabulary similar to the approach described in Section 3.2 to ensure an even number of collisions. The Superbloom architecture is chosen to have a comparable model size to the baseline BERT model.

We compare four models: For the non-hashed baselines, we have a large model with embedding dimension $d = 256$, and a small model with $d = 64$. And we have two Superbloom models with similar model sizes. We list the parameters in Table 4. In Table 5 we list the recall metrics for the

| model | $\alpha$ | $d$ | $n_A$ | $d_F$ | $L$ | #parameters |
|---|---|---|---|---|---|---|
| baseline-h64 | 1 | 64 | 4 | 256 | 12 | 62.6M |
| baseline-h256 | 1 | 256 | 8 | 1024 | 12 | 254.4M |
| hash40-h512 | 40 | 512 | 8 | 2048 | 12 | 62.3M |
| hash20-h1024 | 20 | 1024 | 16 | 4096 | 12 | 246.3M |

Table 4: The model parameters.

models. We observe that with comparable model size, Superbloom outperforms the baseline model in all the recall metrics, and the improvement is more significant for smaller model size.

| model name | rec@1 | rec@10 | rec@20 | model name | rec@1 | rec@10 | rec@20 |
|---|---|---|---|---|---|---|---|
| baseline-h64 | 28.4% | 44.9% | 48.6% | baseline-h256 | 37.2% | 57.4% | 63.3% |
| hash40-h512 | 31.7% | 48.3% | 52.9% | hash20-h1024 | 39.2% | 58.5% | 64.5% |

Table 5: Recall metrics.

Since many named entities are included in the larger vocabulary, the Superbloom model shows that it may have better "understanding" or representation of those entities. We show some anecdotal evidence in Appendix E by comparing predictions of pretrained BERT and Superbloom model on some fill-in-the-blanks examples. The BERT model often predicts generic words, seemingly ignoring other named entities in the sentence. The Superbloom model, on the other hand, can often fill in the blank with related entities.

## 5 CONCLUSION

Our experiments show that the multi-layer Transformer is effective for achieving high accuracy on hashed inputs, represented using Bloom filter digests. Besides applying it to tasks with large vocabularies, it also points to a few interesting future research directions.

The Transformer model has been mostly studied in natural language settings and for sequence data. In our setup, we show that it can work effectively with sets of hashed entities. We hope that by investigating this simpler setup, it can help us better understand the properties of the Transformer. For example, due to hashing, each token is similar to words with multiple meanings, so its embedding can be viewed as a combination, possibly linear (Arora et al., 2018), of the embeddings of multiple entities. A multi-layer Transformer model may provide a mechanism for iteratively filtering such noisy representations, using the context. It would be interesting to further study this mechanism.

While hashing adds noise to the learned representations, it can also increase the flexibility of these representations – when we hash multiple entities to the same token, the model is free to allocate the corresponding embedding unevenly among entities, which results in a different effective embedding dimension for each entity. Such learned capacity allocation might be more efficient than using a fixed embedding size or frequency-based allocation. Of course, an effective "denoising" model is a pre-requisite for such an approach to work. Perhaps Superbloom, with its strong denoising ability, can help further realize the potential of embedding models on hashed vocabularies.

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

## A  TRANSFORMER ARCHITECTURE

We briefly recall the Transformer architecture following Vaswani et al. (2017). Each transformer layer is a function $T : \mathbb{R}^{mn \times d} \to \mathbb{R}^{mn \times d}$ which transforms a sequence of $mn$ embeddings[8] in $\mathbb{R}^d$ to another sequence of $mn$ embeddings in the same space. Identifying the sequence $x_{i,j}$ with the matrix $X \in \mathbb{R}^{mn \times d}$, we can write $T$ as the composition $T = F \circ A$, where

- $A$ is an attention function,

$$A(X) = \sum_{a=1}^{n_A} \sigma\left((XQ_a)(XK_a)^\top / \sqrt{d_A}\right) XV_a W_a^\top, \tag{1}$$

where $a \in \{1, \ldots, n_A\}$ indexes attention heads, $d_A \le d$ is an internal embedding dimension (usually $d_A = d/n_A$), and for each $a$, $Q_a, K_a, V_a, W_a \in \mathbb{R}^{d \times d_A}$. Finally, $\sigma : \mathbb{R}^{n \times n} \to \Delta([n])^n$ is the row-wise softmax function, given by

$$\sigma(Y)_{ij} = \frac{\exp(Y_{ij})}{\sum_{l=1}^n \exp(Y_{il})}. \tag{2}$$

One interpretation of the attention function is that it forms the $i$-th output embedding by taking a convex combination of input embeddings weighted by the softmax weights, followed by a low-rank transformation $V_a W_a^\top \in \mathbb{R}^{d \times d}$.
- $F$ is a fully connected feed-forward network given by $F(X) = \mathrm{ReLU}(XU_1 + b_1)U_2^\top + b_2$, where $U_i \in \mathbb{R}^{d \times d_F}$ for some $d_F \ge d$.

A residual connection and layer normalization are also applied at each stage $A, F$.

## B  THE QUALITY OF BEAM SEARCH

We investigate the effect of beam search width (parameter $B$ in Algorithm 1) on model quality. Table 6 shows rec@$k$ for $k = 1, 10, 20$ for different beam widths $B$, using a small number of test examples, for the Superbloom model described in Section 3.2. In all our experiments, we run approximate inference with one step.

We observe that while the quality generally increases with an increased beam width, the increase in rec@$k$ is only marginal when $B \ge k$. Thus, to obtain highest rec@$k$, it is sufficient to set the beam width to $B = k$.

| beam width | rec@1 | rec@10 | rec@20 |
|---|---|---|---|
| B=1 | 53.0% | 56.0% | 56.0% |
| B=10 | 53.2% | 68.2% | 69.1% |
| B=20 | 53.2% | 67.9% | 71.0% |
| B=100 | 53.2% | 67.8% | 71.5% |

Table 6: Recall metrics at different beam width.

## C  BEAM SEARCH WITH GENERAL SCORE FUNCTIONS

The beam search algorithm described in Algorithm 1 can be generalized to any ranking function that is increasing, in the following sense.

The output of the model defines, for each candidate $s \in \mathcal{S}^O$, a vector of scores $(p_j(\eta_j(s)))_{j=1,\ldots,m}$. To sort the candidates, one can define an aggregated score $\gamma(s) = c((p_j(\eta_j(s)))_j)$ for any function $c : \mathbb{R}^m \to \mathbb{R}$ that induces a total ordering over $\mathbb{R}^m$. One natural assumption to require of $c$ is that it be increasing, in the sense that if $\rho \succeq \rho'$ element-wise then $c(\rho) \ge c(\rho')$. This holds for

---

[8]A minor difference with the original Transformer model is that we operate on $\mathbb{R}^{mn \times d}$ instead of $\mathbb{R}^{n \times d}$, since we have $m$ embeddings for each element in the sequence.

$c(\rho) = \sum_j \log \rho_j$ (used in Section 2.5), but also includes a much larger class, e.g. $c(\rho) = \min_j \rho_j$ or $c(\rho) = \max_j \rho_j$. The beam search Algorithm 1 can be immediately generalized to any such function. The only required change is to replace the condition on line 7 ($\gamma(s^\star) \geq \sum_j \log(p_j^b)$) with $\gamma(s^\star) \geq c(p^b)$. To prove correctness, one needs to verify that the optimality certificate holds in this general case, i.e.,

**Lemma 1.** *Let* $\gamma(s) = c((p_j(\eta_j(s)))_j)$*, where c is increasing in the sense defined above, and let* $p^b$*,* $S_j^b$ *be as defined in Algorithm 1. Then,*

$$\forall s \notin S^b, \gamma(s) \leq c(p^b).$$

It follows from the lemma that the algorithm can terminate whenever $\gamma(s^\star) \geq c(p^b)$, since $\gamma(s^\star) \geq \gamma(s)$ for all $s \in S^b$ (by definition of $s^\star$) and for all $s \notin S^b$ (by the lemma).

*Proof.* Since $S^b = \cup_j \{s : p_j(\eta_j(s)) \geq p^b\}$, then the complement of $S^b$ is the set $\{s : p_j(\eta_j(s)) < p_j^b \forall j\}$. Thus, since $c$ is increasing, it follows that for all $s \notin S^b$, $c((p_j(\eta_j(s)))_j) \leq c(p^b)$, as claimed. $\square$

## D COMPARISON OF DIFFERENT HASHING SCHEMES

We have used random hashing functions in Superbloom. One natural alternative is "coherent" hashing, in which we map similar entities to the same hash bucket. A potential benefit of coherent hashing is that it may use embedding capacity more effectively by sharing it among similar entities. However, the downside is that it becomes difficult to distinguish those similar entities.

To create a coherent hashing function, we first run a co-occurrence factorization algorithm and then group similar entities together using the following procedure, designed to guarantee equal-sized hash buckets. For each entity, in decreasing frequency order, we compute the nearest neighbors (scored using cosine similarity), then create a hash bucket that includes the elements and its $\alpha - 1$ nearest neighbors which have not been already assigned a bucket. When creating a second coherent hash function, we add the constraint that any pair of elements that share a bucket for the first hash function cannot be assigned to the same bucket in the second hash. This ensures that no two elements have the same collision in both hash functions.

We carry out the experiments on the data set with smaller vocabulary (500K). We train different models that all use two hash functions, with the following configurations: both random, one random and one coherent; and both coherent. We also use different hashing densities $\alpha = 10$ and $\alpha = 20$. All the models have the same hyper-parameters as the superbloom-l12 model in Section 3.4. The results are given in the following table.

| model | $\alpha$ | #coherent hashing | token rec@1 | entity rec@1 |
|-------|----------|-------------------|-------------|--------------|
| hash10-00 | 10 | 0 | 36.32% | 52.50% |
| hash10-01 | 10 | 1 | 38.19% | 50.20% |
| hash10-11 | 10 | 2 | 38.55% | 34.70% |
| hash20-00 | 20 | 0 | 33.39% | 43.70% |
| hash20-01 | 20 | 1 | 36.98% | 41.10% |
| hash20-11 | 20 | 2 | 37.65% | 30.20% |

Table 7: Random hashing versus coherent hashing.

We observe that with coherent hashing, we get higher accuracy for predicting hash tokens but lower accuracy for predicting entities. And the entity recall@1 is significantly lower when both hash functions are coherent. This indicates that with higher coherence, it becomes increasingly difficult for the model to make finer distinctions between similar items.

# E  EXAMPLES OF WIKIPEDIA ENTITY PREDICTIONS

1. Examples of pairing "Copenhagen" with different entities. The predictions vary according to the context, from Danish cities, to major European cities, to Danish royalty, and Danish culture. There is a one unrelated result (underlined), which disappears in the presence of additional context.

Copenhagen [MASK]
Denmark Oslo Stockholm Paris Berlin Aarhus Danish_language University_of_Copenhagen Sweden Copenhagen

Copenhagen Aarhus [MASK]
Denmark Odense Copenhagen Aalborg Aarhus Oslo Malmö Max_Wilms Stockholm Esbjerg

Copenhagen Paris [MASK]
Berlin Denmark London Oslo Rome Vienna Stockholm New_York_City Brussels Hamburg

Copenhagen Dynasty [MASK]
Denmark Margrethe_II_of_Denmark Danish_language Copenhagen Catholic_Church Rome Christian_V_of_Denmark Jutland When_We_Wake_Up Frederik,_Crown_Prince_of_Denmark

Copenhagen Dynasty Danish_language [MASK]
Denmark     German_language     Margrethe_II_of_Denmark     Catholic_Church     Copenhagen     English_language     Princess_Benedikte_of_Denmark     Danish_language     Frederik,_Crown_Prince_of_Denmark Christian_V_of_Denmark

2. Examples of Jazz musicians. These relatively long and rare name entities would not appear in the vocabulary of a word piece model.

Miles_Davis [MASK]
Jazz  Columbia_Records  Miles_Davis  John_Coltrane  Dizzy_Gillespie  Bill_Evans  Album Sonny_Rollins AllMusic Charles_Mingus

John_Coltrane [MASK]
Miles_Davis  AllMusic  Jazz  A_Love_Supreme  Rolling_Stone  Elvin_Jones  Albert_Ayler Tenor_saxophone New_York_City Drum_kit

Miles_Davis John_Coltrane [MASK]
Jazz  Charles_Mingus  Album  AllMusic  Miles_Davis  Dizzy_Gillespie  Thelonious_Monk Sonny_Rollins Charlie_Parker Bill_Evans

3. Example showing that the prediction is the set union if two entities are not related.

Miles_Davis Thailand [MASK]
Vietnam Bangkok Japan Miles_Davis Cambodia Malaysia Jazz Indonesia Thai_language Brazil Myanmar Rock_music Dizzy_Gillespie John_Coltrane

4. Examples for completing location analogy task!

Texas Austin,_Texas Florida [MASK]
Miami   Houston   Orlando,_Florida   Dallas   Jacksonville,_Florida   Fort_Lauderdale,_Florida Tampa,_Florida Georgia_(U.S._state) Tallahassee,_Florida St._Petersburg,_Florida

Tunisia Tunis Thailand [MASK]
Bangkok Philippines Montcau Tokyo Malaysia Singapore Indonesia Pattaya Vietnam Thai_language

Figure 4: Examples of Superbloom model predictions. For each example, we output the top 10 predictions of the model (computed using Algorithm 1 with a beam width $B = 10$). The entity names shown here are obtained by removing the prefix "https://en.wikipedia.org/wiki/" from the entity URL.

## F  EXAMPLES OF NATURAL LANGUAGE ENTITY PREDICTIONS

---

Miles Davis is a Jazz musician, he is similar to [MASK].

**BERT:** jazz himself beethoven him davis chopin bowie williams jones
**baseline-h256:**  miles_davis john_coltrane bill_evans charlie_parker louis_armstrong sonny_rollins keith_jarrett thelonious_monk jazz duke_ellington
**hash20-h1024:** miles_davis john_coltrane charlie_parker thelonious_monk dizzy_gillespie bill_evans billie_holiday duke_ellington humans_is louis_armstrong

Empire state building is an iconic site of [MASK1] , it is close to [MASK2] .

[MASK1]
**BERT:** architecture chicago manhattan downtown pittsburgh art philadelphia history washington america
**baseline-h256:** architecture modern_art contemporary_art modern_architecture national_significance new_york art its_day historical_significance the_city
**hash20-h1024:**  the_city new_york lower_manhattan manhattan the_neighborhood downtown wall_street the_area harlem architecture

[MASK2]
**BERT:** downtown it chicago philadelphia rome london broadway manhattan chinatown campus
**baseline-h256:**  downtown downtown_pittsburgh city_hall new_york the_city times_square columbia_university san_francisco philadelphia the_pentagon
**hash20-h1024:**  central_park city_hall times_square wall_street union_station broadway lower_manhattan the_pentagon fifth_avenue carnegie_hall

---

Figure 5: Natural language fill-in-the-blank examples. BERT is the base BERT model in Devlin et al. (2018); baseline-h256 and hash20-h1024 are the Superbloom models with 1M vocabulary, with model parameters listed in Table 4.

