# OpenReview forum: "Superbloom: Bloom filter meets Transformer"
_ICLR.cc/2020/Conference — Reject_

### Official Review · AnonReviewer2 · 2019-10-22
**Official Blind Review #2**

**Rating:** 6

**Review:**

This work presents Superbloom, which applies the bloom filter to the Transformer learning to deal with large opaque ids. Quantitative results demonstrate it can be efficiently trained and outperforms non-hashed models of a similar size. The authors also highlight an important distinction from learning using the entire vocabulary: the depth is crucial to get good performance.

The size of the vocabulary could be largely reduced through hashing, which makes a larger embedding dimension and more complex internal transformation eligible and thus better performance. To make the story complete, it is good to have the results with the same embedding dimension and internal complexity as the baseline model to see the limitations. In Section 4, it is equally interesting to compare the performance between the bloom filter reduction and the data-driven word piece reduction. That is, models learned with the bloom filter applied to the whole vocabulary and models learned with the word pieces only.

I think the empirical contribution is above the bar, but I do not think the authors gave enough credit to (Serra & Karatzoglou, 2017). The adoption of bloom filter on large opaque ids has already been proposed in (Serra & Karatzoglou, 2017) as bloom embeddings to deal with sparse high-dimensional binary-coded instances. It seems that the technical part of Superbloom, including the hashing and the inference, is the same as those in (Serra & Karatzoglou, 2017). I would appreciate a lot if the authors could revise the presentation to either emphasizing the adoption of the work, or highlighting the differences.



**Experience Assessment:**

I have read many papers in this area.

**Review Assessment: Checking Correctness Of Derivations And Theory:**

N/A

**Review Assessment: Checking Correctness Of Experiments:**

I carefully checked the experiments.

**Review Assessment: Thoroughness In Paper Reading:**

I read the paper thoroughly.

---

> ### Author Response · Authors · 2019-11-15
> **Authors response**
>
> Thanks for your suggestions. We have revised the paper (in Section 1) to emphasize that the idea of multi-hashing was first proposed in (Serra & Karatzoglou, 2017), and our contribution is in demonstrating the effectiveness of the Transformer model on Bloom filter digests and an efficient approximate inference algorithm it enables (described in Section 2.5).

---

> > ### Comment · AnonReviewer2 · 2019-11-15
> > **Response received**
> >
> > Thanks for the detailed response.

---

### Official Review · AnonReviewer3 · 2019-10-23
**Official Blind Review #3**

**Rating:** 3

**Review:**

The authors propose to learn a Transformer in embedded spaces defined via $m$ hash functions, with application to (fixed length) sequence-to-sequence classification for NLP. The method differentiates itself, in large part, by formulating model outputs in terms of $m$ distinct softmax layers, each corresponding to a hash of output space $\mathcal{Y}$. The relative low dimensionality of each hashed output space $d \ll \vert\mathcal{Y}\vert$ helps to avoid computational bottlenecks associated with wide softmax layers.

For a given labelled pair $(\mathbf{x}, y)$, the model outputs a matrix $\mathbf{P} \in \mathbb{R}^{m \times d}$, whose $i$-th row is trained using the $i$-th hash $h_{i}(y)$. At test-time, predictions are made by finding tokens $y^{*} \in \mathcal{Y}$ whose hashes best 'align' with model outputs $\mathbf{P}$. By way of example, target alignment might be defined in terms of the aggregated log likelihood

    $\ell(y; \mathbf{P}) = \sum_{i=1}^{m} \log(p_{i, h_{i}(y)})$,

where $p_{i, h_{i}(y)}$ can be understood as the element of $\mathbf{P}$ corresponding to the $i$-th hash of token $y$.

At this time, the work is significantly hindered by a lack of clarity. This issue begins with the chosen notation, which routinely obscures otherwise simple points. Similarly, the section on test-time predictions (Sect 2.5) is needlessly hard to follow. Given its pivotal role, this section feels strangely rushed. Some unanswered questions I had include:
  a) What happens if two tokens' hashes collide all $m$ times?
  b) How were hash functions inverted and what was the cost of doing so?
  c) How does the test-time throughput of the proposed compare with that of alternatives?


Questions:
  - Did you compare against different approaches to sparse softmax, such as LSH-based methods [1]?
  - What was the impact of approximate vs. exact inference and which was used during experiments?
  - How important were embedding matrices $E^{I}, E^{O}$? What happens if you directly feed hashes into the Transformer or use random projections for $E$?.


[1] "Deep Networks With Large Output Spaces", Vijayanarasimhan et al, 2015

**Experience Assessment:**

I have read many papers in this area.

**Review Assessment: Checking Correctness Of Derivations And Theory:**

I assessed the sensibility of the derivations and theory.

**Review Assessment: Checking Correctness Of Experiments:**

I assessed the sensibility of the experiments.

**Review Assessment: Thoroughness In Paper Reading:**

I read the paper at least twice and used my best judgement in assessing the paper.

---

> ### Author Response · Authors · 2019-11-15
> **Authors response**
>
> Thank you for your comments. Please see our inline response.
>
> > At this time, the work is significantly hindered by a lack of clarity. This issue begins with the chosen notation, which routinely obscures otherwise simple points. Similarly, the section on test-time predictions (Sect 2.5) is needlessly hard to follow. Given its pivotal role, this section feels strangely rushed.
>
> We made several updates to Section 2 to simplify the notation. We highlight the following related to your comments. We rewrote Section 2.5 for the special case of maximizing the sum of log likelihoods. This was done for two reasons: working with a special case allowed us to simplify the notation/discussion, and the same function was used in the related work on Bloom embeddings (Serra & Karatzoglou, 2017), where exhaustive scoring is used. We moved the more general case to the appendix.
>
> > Some unanswered questions I had include:
> >  a) What happens if two tokens' hashes collide all  times?
>
> Indeed, in the case of complete collision, one cannot distinguish the original tokens. Thank you for pointing this out. In fact, we store the hash maps and their inverse explicitly as look-up tables, so we can make sure this does not happen (by regenerating the hashes if we detect complete collisions, which happens rarely for our choice of parameters). We added more description about it in Section 2.1.
>
> > b) How were hash functions inverted and what was the cost of doing so?
>
> We store inverse look-up tables, as mentioned above. We updated the paper to explicitly mention this. We also added a discussion of the added memory cost of storing these tables and why this is negligible (it is dominated by the embedding tables).
>
> > c) How does the test-time throughput of the proposed compare with that of alternatives?
>
> The time breaks down into two parts: the transformer encoding and the nearest neighbor search. The former should be common to different models. With Superbloom, the nearest neighbor search is much more efficient, as the vocabulary size is reduced by many times.
>
> > Questions:
> >  - Did you compare against different approaches to sparse softmax, such as LSH-based methods [1]?
>
> We compared to sampled softmax in the paper. But which softmax strategy to use is somewhat orthogonal to our work: one can still apply different softmax approximation on top of our model. However, since Superbloom reduces the vocabulary size to sufficiently small size, we can now afford to run full softmax -- such an approach has been established as a better alternative than sampled or approximate softmax through work such as NMT and BERT. Accuracy wise, BERT model has much higher accuracy than [1] (~70% on word pieces prediction vs <2% recall@1) In our work, we obtain accuracy of recall@1 of 30%+ (Table 5) for uni/bigram predictions. Such accuracy is a combination of more powerful Transformer model as well as the full softmax loss (important for recall@k for small k’s).
>
> >  - What was the impact of approximate vs. exact inference and which was used during experiments?
>
> Appendix B presents the experiments on the effect of the beam search width on model quality. We find that, if the quality is measured using recall@k, the quality improves significantly by increasing the width B up to k, and only marginally beyond k.
>
> >  - How important were embedding matrices ? What happens if you directly feed hashes into the Transformer or use random projections for ?.
>
> It is an interesting idea to push all the embedding into the Transformer model. That’s interesting future work, which we alluded to in the conclusion (Section 5).

---

### Official Review · AnonReviewer1 · 2019-10-25
**Official Blind Review #1**

**Rating:** 3

**Review:**

The paper proposes to use codes based on multiple hashing functions to reduce the input and output dimensions of transformer networks. The novelty of the idea is mostly to integrate these codes with transformers. The authors present experiments on two tasks showing that keeping overall model size fixed (i..e, number of parameters), transformers with shorter (but denser) binary codes achieve better performances than standard transformers using one-hot encodings. The gain mostly comes from using larger embedding sizes and larger intermediate layers.

While the technical contribution is limited, because most of the principles are already known or straightforward, the main contribution of the paper is to show that random hash functions are sufficient to create significantly shorter codes and maintain good performances. Such codes provide more freedom in terms of where to put model capacity (larger embeddings, more transformer layers, etc.) which may be useful in applications where most of the model parameters are in embedding matrices.

The value of such a paper resides mostly in the experimental study. On the bright side, the experiments present in sufficient details the impact of the various hyper parameters and the new trade-offs model size/performance that can be achieved. On the other hand, the experiments are carried out on non-standard tasks without previously published baselines, and it is unclear why. Since the method is applicable to any problem involving natural language data (and more generally categorical values, such as knowledge base completion), I would have expected experiments on tasks with a well-defined state-of-the-art. This makes the experiments in the paper look more like "proofs of concept", and they are less convincing than they should be.

detailed comments:
- The paper really is *not* about bloom filters (Bloom filters are data structures that represent sets and efficiently answer membership queries). It is about using codes of identifiers of lower dimension than one-hot encoding. This idea has been used in multi class classification setting (i.e., for the output layer) since (at least) Dietterich & Bakiri (1995) "Solving Multiclass Learning Problems via Error-Correcting Output Codes" (with more insights on what makes a good code for prediction problems). The authors borrow from Bloom filter the way to create the codes using random hash functions, but the analogy stops here.

- following the comment above, and assuming I understood correctly: There is an originality on the paper compared to other works that use binary codes/bloom filters: In the current paper, the authors actually predict the result of individual hashing functions. This is different from predicting the binary encoding that results from using the "or" of one-hot encodings generated by several hash functions, as would be done in approaches (truely) based on Bloom filters. For instance, if there are m hash functions taking values in {1, ..., P}, an approach based on Bloom filters would predict a binary output of dimension P, while here there are m multiclass problems with P classes (IIUC). This difference from previous work may be significant in practice.

- I found the description of the link prediction task (section 3.1) rather cryptic:
* "from each page, we take a random contiguous segment of entities". If I understand clearly, the text is filtered out and only links are kept (?). Links are replaced by the entity id they point to. What happens to the entity the page is about? Is it added at the start?
* 'For evaluation, we hold out one random entity from a random segment on a test page. ": what does "holding out" mean? From my understanding, it means replaced by the [MASK] token, but it could also mean removed altogether from the input sequence.
* the task is to predict masked links in sequences of links with the surrounding text filtered out. Does that correspond to any real-life prediction problem (i don't see which one)? Is this "task" intended to serve as unsupervised pre-training of embeddings? If yes, maybe the authors might say so and give example applications.

- For the natural language task:
* ", then apply a Bloom filter to reduce the vocabulary size." -> From my understanding, there is no bloom filter here. If I understand, what is done is to represent unigrams and bigrams by their bloom digest to reduce the input dimension.

- "We hold out 10% random entity pages for testing. " -> is there a validation set? How do you choose the hyper parameters?


minor comments:
- "Since the Wikipedia site usually removes duplicates of links on each page, the distribution of pages is rather long tail." -> I wouldn't be surprised if the distribution was long tailed even without this specific policy
- I found the formalization/notation more confusing than helping because it is not really thorough (there is no distinction between sets and sequences, "1[\eta_j(t_i)] ... where 1[.] is the indicator function" -> what is the "indicator function" of a number?)

**Experience Assessment:**

I have published one or two papers in this area.

**Review Assessment: Checking Correctness Of Derivations And Theory:**

I carefully checked the derivations and theory.

**Review Assessment: Checking Correctness Of Experiments:**

I assessed the sensibility of the experiments.

**Review Assessment: Thoroughness In Paper Reading:**

I read the paper at least twice and used my best judgement in assessing the paper.

---

> ### Author Response · Authors · 2019-11-15
> **Authors response**
>
> We thank the reviewer for the detailed comments. Please see our inline responses.
>
> detailed comments:
> > - The paper really is *not* about bloom filters ...
> > - following the comment above, and assuming I understood correctly: There is an originality on the paper compared to other works that use binary codes/bloom filters...
>
> Thank you very much for the connection to error-correcting output codes as well as the comments on the connection to Bloom filter. The k-hot representation of the input is identical to Bloom filter, but you are right that the output prediction is not on the individual bits in the Bloom filter, but rather on the indices of the non-zero bits, which is formulated as a multi-class classification problem. So our output prediction task reduces a S-way classification to m H-way (using the notation in our paper) classification, instead of going all the way to binary classifications as in ECOC models. This is probably important for obtaining high accuracy. We have revised the paper to discuss this connection in Section 1 and added footnote 4 to emphasize this difference.
>
> > - I found the description of the link prediction task (section 3.1) rather cryptic:
> > * "from each page, we take a random contiguous segment of entities". If I understand clearly, the text is filtered out and only links are kept (?). Links are replaced by the entity id they point to. What happens to the entity the page is about? Is it added at the start?
>
> Yes, this is the setup. The entity page is not used. All the semantic relationship is learned from the co-occurrence of links on Wikipedia pages.
>
> > * 'For evaluation, we hold out one random entity from a random segment on a test page. ": what does "holding out" mean? From my understanding, it means replaced by the [MASK] token, but it could also mean removed altogether from the input sequence.
>
> Assume that the set of entities is {A,B,C,D}, we remove one of them, e.g., “C” and ask the inference to predict what is missing from the set {A,B,D}. As this is a transformer, to do inference, we add a mask token to the set {A,B,D,[MASK]}, the prediction of mask is compared to the withheld “C”. We added clarification in Section 3.1.
>
> > * the task is to predict masked links in sequences of links with the surrounding text filtered out. Does that correspond to any real-life prediction problem (i don't see which one)? Is this "task" intended to serve as unsupervised pre-training of embeddings? If yes, maybe the authors might say so and give example applications.
>
> The main objective of the paper is to present a technique that is useful for reducing vocabulary size. We focus on the pretraining task, i.e. masked language model task, to carry out our study. In natural language understanding, such task is usually treated as the pretraining task but is also commonly used as the yardstick for the quality of the model (e.g. the commonly used perplexity metric). In some other domains, for example recommender systems, it can be directly used for making recommendations by using the prediction vector to retrieve candidate items. The large vocabulary size is more common in those tasks.
>
> > - For the natural language task:
> > * ", then apply a Bloom filter to reduce the vocabulary size." -> From my understanding, there is no bloom filter here. If I understand, what is done is to represent unigrams and bigrams by their bloom digest to reduce the input dimension.
>
> We use Bloom filter to refer to the multi-hashing scheme and to the set representation of the input -- the k-hot representation of the input (as is done in our model) is a faithful Bloom filter representation but the output prediction is not --- as discussed above, the predictions are not on the bits (they would be too sparse) but rather as a multi-class classification task. We have added discussion in our paper.
>
> > - "We hold out 10% random entity pages for testing. " -> is there a validation set? How do you choose the hyper parameters?
>
> We only use the training data for all the tuning, using the training metrics. Actually, our model does not seem to overfit, probably due to large data size.
>
> > I found the formalization/notation more confusing than helping because it is not really thorough (there is no distinction between sets and sequences, "1[\eta_j(t_i)] ... where 1[.] is the indicator function" -> what is the "indicator function" of a number?)
>
> Thank you for these suggestions. We made several updates to the notation, including removing the indicator function (which was only used to map an element to the one-hot encoding of that element) and using sequences throughout.

---

### Author Response · Authors · 2019-11-15
**Summary of revision**

We thank the reviewers for their comments and suggestions. We would like to emphasize that our main contribution is to recognize the synergy between the multi-layer Transformer model and the Bloom filter (as the title suggests). While multiple hashing has been proposed before for both input and output representation, we demonstrate that its potential is more fully realized by multi-layer Transformer models, with their ability to remove "hashing noise" using the context.

In the revision, as suggested by the reviewers, we connected and contrasted our contribution better to previous work and also improved the clarity of our presentation.  We provide a summary below.

- We better recognize the connection of our work to previous work in Section 1. This includes re-iterating that the idea of using hashing was already proposed before and discussing the connection between our prediction task to the error-correcting output code. We thank reviewer 1 for suggesting the connection.
- We emphasized our key insight on the importance of applying a multi-layer Transformer model for Bloom filter digests.
- We simplified the notation in Section 2. For example, we removed matrix notation and the indicator function notation. We simplified the inference algorithm (Section 2.5) for the special case of maximizing the log likelihood. We added a section to the appendix to discuss how this generalizes to other ranking functions, and formally stated in a lemma the property that we rely on for correctness.
- We clarified (in Section 2.5) that we store hashing functions and their inverse in look-up tables.

---

### Decision · Program_Chairs · 2019-12-19

**Decision:**

Reject

**Comment:**

This paper presents to integrate the codes based on multiple hashing functions with Transformer networks to reduce vocabulary sizes in input and output spaces. Compared to non-hashed models, it enables training more complex and powerful models with the same number of overall parameters, thus leads to better performance.
Although the technical contribution is limited considering hash-based approach itself is rather well-known and straightforward, all reviewers agree that some findings in the experiments are interesting. On the cons side, two reviewers were concerned about unclear presentation regarding the details of the method. More importantly, the proposed method is only evaluated on non-standard tasks without comparison to other previous methods. Considering that the main contribution of the paper is in empirical side, I agree it is necessary to evaluate the method on more standard benchmarking tasks in NLP where there should be many other state-of-the-art methods of model compression. For these reasons, I’d like to recommend rejection.